# Causal Mechanisms of Subpolar Gyre Variability in CMIP6 Models

Swinda K.J. Falkena<sup>1</sup>, Henk A. Dijkstra<sup>1,2</sup>, and Anna S. von der Heydt<sup>1,2</sup>

**Correspondence:** Swinda K.J. Falkena (s.k.j.falkena@uu.nl)

Abstract. The subpolar gyre is at risk of crossing a tipping point under future climate change associated with the collapse of deep convection. As such tipping can have significant climate impacts, it is important to understand the mechanisms at play and how they are represented in modern climate models. In this study we use causal inference to investigate the representation of several proposed mechanisms of subpolar gyre variability in CMIP6 models. As expected, an increase in sea surface salinity or a decrease in sea surface temperature leads to an increase in mixed layer depth in nearly all CMIP6 models due to an intensification of deep convection. However, the effect of convection to modify sea surface temperature due to re-stratification is less clear. In most models the deepening of the mixed layer caused by an increase of sea surface salinity, does result in a cooling of the water at intermediate depths. The feedback from the subsurface temperature through density to the strength of the subpolar gyre circulation is more ambiguous, with fewer models indicating a significant link. Those that do show a significant link, do not agree on its sign. The CMIP6 models that have the expected sign for the links from density to the subpolar gyre strength and on to sea surface salinity, are also the models in which abrupt shifts in the subpolar gyre region have been found in climate change scenario runs. One model (CESM2) contains all proposed mechanisms, with both a negative and delayed positive feedback loop being significant.

#### 1 Introduction

The North Atlantic Subpolar Gyre (SPG) is considered as one of the tipping elements in the climate system (Lenton et al., 2023; Loriani et al., 2023). SPG tipping refers to a decadal (or longer) shutdown of convection in the interior of the SPG and a drastic weakening of the baroclinic part of the SPG circulation. Support for such tipping behaviour comes from paleoclimate reconstructions, in particular from high resolution bivalve data from the North Icelandic shelf. These data indicate two episodes of SPG tipping prior to the Little ice Age (Arellano-Nava et al., 2022), where both events appear to be driven by freshwater input, for example through the melting of the Greenland Ice Sheet during the preceding Medieval climate optimum. A temporary shutdown of convection has also been observed during the Great Salinity Anomalies in the historical period, but these lasted only a few years before convection did restart (Gelderloos et al., 2012; Kim et al., 2021).

Multiple studies have shown the possibility of an abrupt change in (sea) surface temperature in the SPG region in CMIP models (Sgubin et al., 2017; Swingedouw et al., 2021), which in several models coincides with a reduction in the mixed layer depth (and thus convection). More recently, a collapse of convection and corresponding cooling in the subpolar North Atlantic

<sup>&</sup>lt;sup>1</sup>Department of Physics, Institute for Marine and Atmospheric research Utrecht (IMAU), Utrecht University, Utrecht, the Netherlands

<sup>&</sup>lt;sup>2</sup>Centre for Complex Systems Studies, Utrecht University, Utrecht, the Netherlands

was found in CESM2 large ensemble scenario simulations of the near future (Gu et al., 2024). These model studies show the possibility of abrupt temperature shifts in the SPG region, likely already at low levels of global warming, with substantial climate impacts. It is hence of key importance to understand the mechanisms responsible for the temperature shifts in these models and in observations.

30

55

The physical processes determining the SPG are quite well known. The baroclinic part of the SPG circulation consists of an alongshore flow driven by a cross-shore density gradient between the buoyant boundary current and the dense interiors of the SPG's marginal seas (i.e. the Labrador and Irminger Seas). On an annual mean timescale, the buoyancy loss to the atmosphere in these interiors is balanced by buoyancy gain through lateral eddy fluxes originating from the boundary current (Spall, 2004). Higher salinities in the boundary current result in stronger lateral eddy transport and therefore more vigorous convection in the interior. This cools the water column at mid-depth, which strengthens the cross-shore density gradient and hence the baroclinic flow, which in turn increases the shedding of eddies from the boundary current enhancing the lateral transport in a positive feedback (De Steur et al., 2018; Holliday et al., 2020). Also more heat is transported by the eddies, leading to a negative feedback, but these temperature anomalies are quickly damped by the atmosphere. The box model by Born and Stocker (2014) captures these processes in a simplified way and shows that convective and non-convective states, and hence bistability, can exist for the forcing over a substantial part of the parameter space.

However, in CMIP6 models the boundary currents in the SPG's marginal seas are not well represented and the eddy fluxes of heat and salt are highly parameterised. CMIP6 models in addition have biases in their representation of convection, with convection happening too often, too deep, and not in the right location in many models (Heuzé, 2021). Hence, it is not guaranteed that transitions such as in Born and Stocker (2014) could be captured in these models and if not, how temperature shifts associated with changes in SPG convection are caused in some of the models (Sgubin et al., 2017; Swingedouw et al., 2021).

In CMIP6 models, deep convection is driven locally by vertical density differences and occurs when the surface density exceeds that of the waters below. Density is controlled by both temperature and salinity, where either an increase in sea surface salinity (SSS) or a decrease in sea surface temperature (SST) can lead to convection, deepening the mixed layer depth (MLD). These two causes can be represented as in Figure 1 by the orange and brown (A<sub>1</sub>) arrows in the yellow box. It is still a topic of debate which of the two (SST or SSS) is more important for driving variability in convection, and the results differ depending on the timescales of interest. Some studies indicate SSS as the dominant driver of convection (e.g. Hátún et al., 2005; Gelderloos et al., 2012; Yamamoto et al., 2020), where changes in salinity are primarily linked to ocean transport. Others show the importance of surface heat fluxes, which affect SST, on the strength of convection (e.g. Yashayaev and Loder, 2016; Piron et al., 2017). On interannual timescales, SSS is a stronger driver than SST (e.g. Yamamoto et al., 2020).

Convection also feeds back to the surface as warmer water from depth is mixed with the surface water, lowering the surface density. The effect of this re-stratification on temperature has been found to be significant, whereas the effect on salinity is not as strong (Lazier et al., 2002). This can lead to bistability of convective and non-convective states as shown in conceptual models (Welander, 1982) and is represented by the orange and pink  $(A_2)$  arrows in Figure 1. Even without detailed marginal sea processes representations in CMIP6 models, the density in the SPG region affects the strength of the (horizontal) gyre

circulation, which in turn alters SST and SSS. This way changes in convection occurring can alter the strength of the SPG circulation, providing a possible feedback loop (Born and Mignot, 2012).

According to this mechanism, depicted by the large feedback loop (B) in Figure 1, a stronger gyre circulation leads to a reduction of salinity in the SPG centre on short timescales (up to 2 years), but an increase on longer timescales (5 to 10 years) due to eddy transport. An increase of sea surface salinity leads to convection and thus an increase in mixed layer depth on timescales of up to two years. This mixing brings cold surface water down, reducing the temperature of the intermediate water column and increasing the density at depth. In Born and Mignot (2012) the link from MLD to the temperature of the intermediate water column has a lag of around 7 years, where the reason given for the lag is thermal inertia of the water column. This suggests that the effect itself likely occurs on shorter timescales and is sustained by the memory of the water column. Therefore, we refrain from specifying its lag in Figure 1 and hypothesize it is shorter than 7 years. Lastly, the increase of density in the centre of the gyre leads to a strengthening of the circulation through thermal wind balance. Taking the links together, the mechanism, represented by blue, orange, green, red and purple arrows in Figure 1, consists of a negative feedback loop on short timescales and a positive one on longer timescales.

Figure 1. In the yellow box two mechanisms for the interaction between SST, SSS and MLD are shown. Mechanism  $(A_1)$  represents both SSS and SST impacting MLD, whereas mechanism  $(A_2)$  has SSS as the main driver of MLD which in turn feeds back to SST. In the larger loop (B) the mechanism leading to bistability of the subpolar gyre following Born and Mignot (2012) is depicted. The dashed box around SST indicates it is not part of the mechanism (B). The arrows indicate a directional link between two variables, with a solid line indicating a positive effect and a dashed line a negative one. For most links the feedback is expected to be relatively fast, either instantaneous or with a lag up to 2 years. The delayed positive feedback from SPG to SSS is due to eddy transport and is expected to have a lag of 5 to 10 years.

The aim of this study is to investigate the mechanisms of SPG tipping behaviour in CMIP6 models using a causal analysis. The causal framework (Runge, 2018) allows to account for possible confounding factors and the effect of memory. The starting point of this analysis are the three potential mechanisms as shown in Figure 1. In the next section we start with a description of the CMIP6 data that is used, followed by a description of the causal methodology. In Sections 3 and 4 the results are discussed, showcasing how different models perform in representing the different mechanisms ((A) and (B) respectively). We end the paper with a discussion of the results and an outlook.

# 2 Data and Methods

To verify the hypothesized mechanisms in CMIP6 models, two things are required. Firstly, indices for each of the variables shown in Figure 1 are required, which is discussed in Section 2.1. Secondly, we need a method to identify the mechanism describing the interactions between these indices. For this, causal inference is used, which is discussed in Section 2.2.

# **2.1** Data

100

To test for the presence of a mechanism in CMIP6 data, it is desirable that as little confounding or forcing mechanisms are present. Therefore, the piControl runs of CMIP6 are considered (Eyring et al., 2016). The variables of interest in the CMIP6 database are the sea surface salinity (sos), sea surface temperature (tos), mixed layer depth (mlotst), subsurface potential temperature (thetao) and the barotropic streamfunction (msftbarot). The sea surface temperature and sea surface salinity are used to compute the density following the TEOS-10 equation of state (Roquet et al., 2015). The barotropic streamfunction is used as a measure of the strength of the subpolar gyre because it is hard to determine the baroclinic part of the flow and the use of density variables to this end would build in a link between density and the gyre strength. With these five variables we compute (scalar) time series for all six feedback elements shown in Figure 1. The regions used for this computation are detailed in the following paragraphs. For the temperature at intermediate depth we take into account depth levels between 50m and 1000m (as in Born and Mignot (2012)). From hereon we use SPG for the strength of the subpolar gyre, SSS for the sea surface salinity, SST for the sea surface temperature, MLD for the mixed layer depth, SubT for the temperature at intermediate depth and Rho for the density. For the mechanisms (A) linked to convection we consider all 47 CMIP6 models for which at least 100 years of data is available (list in Figure 2). Not all of these models have the barotropic streamfunction available as a variable and thus we are left with 32 models for the analysis of mechanism (B) (see Tab. S1 for the list).

Because of the colder temperatures, and resulting lower surface density, convection primarily occurs in winter (Birol Kara et al., 2003; Heuzé, 2017). Therefore, we focus our analysis on the winter months and compute the mean of each variable over January-February-March. Convection is a local process and the location where it is strongest differs between CMIP6 models, as can be seen in Figure 2. For the initial analysis on the mechanisms of convection (A) we therefore select a different region for each model to compute the SSS, SST and MLD indices. For each model a 5° by 5° box around the location of maximum winter (JFM) mixed layer depth is obtained (restricted to the SPG region as indicated by the dashed line), and the indices are

computed as the spatial average over those boxes. This ensures that for each model we focus on the region where that model has the strongest convection.

For the analysis of the Born hypothesis (B) one of the relevant variables is the barotropic streamfunction, as an indicator of the strength of the subpolar gyre. The interaction between the strength of the circulation and the temperature and salinity of the water differs between locations, since the water flowing into the considered region has a different source and thus different properties. This makes it difficult to compare models when using a separate local box for each model. Therefore, we decide to work with one fixed box for all models when studying this mechanism. The region studied is chosen based on where convection takes place in CMIP6 models. For this we consider MLD as an indicator, assuming deep convection takes place if it exceeds 1000m (Marshall and Schott, 1999). In Figure 2 the number of models in which deep convection takes place during at least 100 model years is shown. It indicates that the Labrador sea is the region in the subpolar gyre where deep convection takes place in most models. Therefore, the region considered in this study is 54-63°N, 47-60°W, indicated by the solid box in Figure 2. The spatial average over this box is computed for all variables considered. Despite not all models showing (deep) convection in this region, this approach does allow for a better comparison of mechanism (B) between models because of the inclusion of the strength of the subpolar gyre. Note that this means that the convection mechanisms (A) are considered local for each model, while mechanism (B) is studied in the whole Labrador Sea area.

# 2.2 Causal Links and Causal Effect

115

To verify whether the convection and Born mechanisms are represented in CMIP6 models we analyse the interaction between variables in two ways for each model. Firstly, causal inference is used to identify whether a connection between two variables is significant (at the 5%-level). Secondly, causal effect determines the strength of each link given a network of links. Both approaches are conducted using the Tigramite package for Python. For the identification of significant links, the Peter and Clark momentary conditional independence (PCMCI) algorithm is used (Runge et al., 2014, 2019b, a). This is a causal discovery algorithm which identifies significant causal links between the input variables for a given set of lag-times. Several versions of the algorithm have been developed, of which we use PCMCI+ because it can give direction to instantaneous (no lag) links (Runge, 2020).

The PCMCI algorithm allows to infer causal links between variables from time series data. Following Pearl et al. (2016) a variable X causes Y if P[Y|do(X=x)] is non-zero, where X is called the parent, Y the child and do(\*) is an intervention. In an experimental setting such as a wave tank, an intervention could be to change the frequency of the forcing to study the effect on wave height. However, in reality such interventions, i.e. do(\*), are not possible and in high-resolution climate models they are computationally too expensive and infeasible. Thus, for the time-series data that is available, a time-lag is used to infer causality, assuming the cause happens prior to the effect. This estimation of the links by the algorithm requires a number of assumptions to be valid, of which a full description is given in Runge (2018). Here we briefly discuss the ones that are most relevant to this study.

Firstly, causal sufficiency is assumed. This means that all causally relevant variables are included and there exist no other (unobserved) variables that influence a pair of variables in the considered set. Secondly, the set of causal links is assumed to

**Figure 2.** The markers indicate the location in the subpolar gyre region (dashed box) where the winter (JFM) mean mixed layer depth has its maximum for each of the 47 CMIP6 models studied. Note that not all markers are visible due to overlapping maxima, mostly because model families have their maxima in the same location (e.g. for CanESM5-1 the location is the same as for CanESM5). The shading shows the number of models (out of the 32 which are used to study the Born hypothesis) in which the mixed layer depth exceeds 1000m in at least 100 model years (out of 500). The black box indicates the region 54-63°N, 47-60°W, chosen as the location where convection takes place in most CMIP6 models.

be stationary, in the sense that the presence and strength of the links does not change in time. The last assumption we mention here is linearity of the links, which strictly speaking is unlikely to be valid. There are multiple metrics that can be used in the algorithm, but the more non-linear they get, the longer it takes to compute, making them unfeasible for our application to many models. Furthermore, linearity has been shown to be a good first order approximation for climate data (Kretschmer et al., 2021; Di Capua et al., 2020). Therefore, we use partial correlation as our metric, extending the commonly used frame of correlation by taking into account the effect of auto-correlation and (included) confounding factors (Saggioro and Shepherd, 2019). To be specific, we use a robust partial correlation measure (RobustParCorr) which is more suitable for non-Gaussian data. This choice is made because especially the mixed layer depth does not follow a Gaussian distribution in most models.

For the second step in the analysis of CMIP6 models the strength of the links in the network is determined. As a measure of the link strength the causal effect from one variable to the other is determined using the causaleffect-class of the Tigramite package, which in its basis performs a conditional regression analysis (Runge, 2021). To compute the causal effect

of variable X on Y, one regresses Y on X conditional on all other parents of Y (given the network). Those other parents are the history of Y itself (as far back as significant), other driving variables Z and possibly the history of X. Given the hypothesized networks as shown in Figure 1, in most cases the conditioning is solely done on the history of both X and Y as far back as found to be significant in the causal inference step. Only for mechanism  $(A_1)$  one also conditions on the other parent (SST or SSS).

# 3 Mechanisms of Convection

We start by studying the two (local) convection mechanisms (A) represented in Figure 1. Because of the close relationship between SSS and SST, causal inference struggles to identify the expected links, especially for instantaneous interaction, where there is a large spread between models (see Fig. S1). For example, in many models a causal link between SST and SSS is identified due to the absence of common drivers, such as the circulation strength of the gyre. Not including these possible drivers means that the assumption of causal sufficiency is violated, making it impossible to robustly identify the presence of links in the models. Therefore, we limit our analysis to the computation of the causal effects between the variables are obtained by using the links for mechanisms (A₁) and (A₂) as shown in Figure 1. We consider both links in (A₁) to be instantaneous, while for (A₂) we take the SSS→MLD link at lag-0 and the MLD→SST link at lag-1 because of the slower effect of re-stratification (see also Fig. S1). Furthermore, we assume a lag-1 memory effect of each of the variables onto itself, which is a feature clearly identified in most models using causal inference. This part of the analysis may be hindered by the violation of causal sufficiency as well, but it can still provide valuable insight into the local interaction mechanisms.

The results for this analysis of causal effects are shown in Figure 3. For mechanism  $(A_1)$  we find that indeed an increase in SSS leads to more convection, i.e. an increase in MLD. A decrease in SST has the same effect, although slightly weaker in most models. The results for mechanism  $(A_2)$  are less pronounced. In most models an increase in SSS causes an increase in MLD, although the effect is weaker than in  $(A_1)$ . The difference with  $(A_1)$  arises because in the computation of the causal effect one conditions on all parents of MLD; in the case of  $(A_1)$  those are SST and MLD at lag-1, whereas in the case of  $(A_2)$  it is only MLD at lag-1. The identified feedback from MLD to SST is not in line with the theory, with most models indicating a negative causal effect. This is likely due to confounding factors (violating causal sufficiency), such as the ocean circulation, that also strongly impact SST (and SSS) and that way obscure the signal. The reason this affects hypothesis  $(A_2)$  and not  $(A_1)$ , is that in  $(A_2)$  SST is a child of SSS (through MLD). In  $(A_1)$  such a relation is absent and thus confounding factors, i.e. parents of both SST and SSS, do not impact the results. Thus, in order to be able to identify a robust and reliable signal we require the hypothesis to be causally sufficient, by which we mean a hypothesis where all relevant variables are included. One hypothesis for SPG variability that is causally sufficient is that of the mechanism (B) shown in the large loop in Figure 1. It includes all variables relevant to bistable variability of the SPG, and thus can be tested using causal inference. This does not mean that the mechanism in reality is not affected by confounding factors, but the hypothesized mechanism is not and thus can be tested. Adding for example only the strength of the subpolar gyre to the local analysis of convection processes discussed here, would

lead to results that are hard to interpret since the location of deepest convection differs strongly between models (as discussed in Section 2.1).

Figure 3. The causal effect for convection mechanisms  $(A_1)$  and  $(A_2)$  as shown in Figure 1. Here both links in  $(A_1)$  as well as the SSS $\rightarrow$ MLD link in  $(A_2)$  are considered at lag-0, while the MLD $\rightarrow$ SST link in  $(A_2)$  is considered at lag-1. The violins give the distribution over all models, with the dots representing individual model values.

# 4 Mechanism of Subpolar Gyre Variability

The verification in CMIP6 models of the hypothesized mechanism (B) as shown in Figure 1 is done in two steps. Firstly, causal inference is used to find which of the links are significant, which is done by studying one link  $X \to Y$  at a time. This means we apply causal discovery to X, Y and the causing variable Z of X ( $Z \to X$ ), accounting for its possible impact on auto-correlation, to verify the significance of the links. It is unlikely that all models agree on which links in the theoretical model are significant at which lags and thus we start with a discussion of the number of models in which the links are found to be significant in Section 4.1. Based on these results we move to the second step; computation of the causal effect. For this we set the lags in the network as found in Section 4.1 and use this to test the strength of the links (Section 4.2). Note that the computations here are done for the variables considered in the Labrador sea box as shown in Figure 2, which is different from the model dependent areas considered before. Comparison between the two regions shows no significant differences in the causal effects for the SSS $\rightarrow$ MLD link in the convection mechanisms (see Fig. S3), indicating this interaction can also be identified when considering the same Labrador Sea region for all models.

# 4.1 Significant Links

The number of models for which each of the five links is significant are shown in Figure 4 for lags up to 10 years. Results for both the theoretical links (bottom row), as well as the influence of variables on themselves (top row), i.e. their memory, are shown. For instantaneous links also the sum of the correctly directed and unknown links is shown. Looking at the top row we see that SPG has a long memory in many models, with significant links of up to 5 years lag (lag-5) in more than 5 models. SSS, MLD and Rho have substantially shorter memory, whereas SubT-memory is two years in most models.

**Figure 4.** The number of models (out of 32) for which a significant link is found (at the 5%-level) for lags of up to 10 years. The top row shows the link of each variable to itself, whereas the bottom row shows the theoretical links. Blue circles indicate the number of directed links following the Born model (Figure 1). For instantaneous links orange triangles show links directed in the opposite direction and both green crosses and red plusses are links for which the direction could not be determined, either because the orientation rules could not be applied or because they conflicted, respectively. Purple squares give the number of models with either a correctly directed or unoriented (unknown) link, i.e. the sum of all links except those in the opposite direction.

For the links of the proposed subpolar gyre variability mechanism (B), we find that the SPG→SSS link is absent in 22 out of the 32 models (either at lag-0 or lag-1), especially when instantaneous causation is concerned (only three models, one of which also has a link at lag-1). At lag-1 eight models show a significant link. The SSS→MLD link is correctly represented in 23 models, either instantaneous or at lag-1 (or both), with 13 models indicating an unknown direction for the instantaneous link. Here, 13 models show a link in the opposite direction (some with a lag-1 link in the right direction), indicating these two variables are strongly connected in models. The instantaneous MLD→SubT link is also found in many models; 22 in this case with half being unoriented. At lag-1, seven models find a significant link. The SubT→Rho link is not present in many models,

with only seven models showing a correctly directed or unoriented instantaneous link and six models indicating a significant link at lag-1. Lastly, the Rho $\rightarrow$ SPG link is found in seven more models than the SPG $\rightarrow$ SSS link, with eleven models (out of the 17) indicating lag-1 to be significant. In total there are six models that have both links with SPG as significant.

All but one model capture at least one step of the process (B). Only for CanESM5-1 no link is found to be significant. This is a model that shows very little convection in the Labrador sea. The best captured links are SSS $\rightarrow$ MLD and MLD $\rightarrow$ SubT, which are found in 23 and 27 CMIP6 models respectively. The links to and from SPG are found in only 17 and 10 models, respectively. This is to be expected as this connection relies on the modelling of advection which is dependent on the used parametrisation (Griffies, Stephen, 2018; Small et al., 2020). The SPG $\rightarrow$ SSS link at longer lag is not found. There is one model, CESM2, that captures all links (except the lagged one), which will be discussed in more detail in the following section.

# 4.2 Causal Effect

To compute the causal effect for the links we need to set the lags at which they are considered. If more than five models indicate a link at a certain lag to be significant, it is included for the computation of the causal effect. This means that for all theoretical links between variables we take into account both instantaneous and lag-1 links, except for SPG $\rightarrow$ SSS where only links at lag-1 are considered. Furthermore, links from SSS, MLD and Rho to themselves at lag-1 are used, while for SubT lag-2 is considered in addition. For SPG lags up to five years are taken into account.

In Figure 5 the causal effect of each link is shown for models with and without that link being significant following the causal discovery step (values for individual models are given in Fig. S5). Distinguishing between significant and non-significant models allows to separate the models in which we do expect to find the relevant dynamics from those where we do not (e.g. because they show little convection in the Labrador Sea). For the instantaneous SSS→MLD link (orange violins) all (significant) models identify the correct sign, with an increase in salinity leading to a deeper mixed layer. This is in line with the results found for the mechanisms of convection found in Section 3. Also for the instantaneous MLD→SubT link (green violins) most significant models identify the right direction with an increase in MLD leading to a cooling of SubT. At lag-1 both the SSS→MLD and MLD→SubT link are found with the opposite sign in most models where they are significant, contrary to the theoretical model. This sign-change is only found when taking into account the longer memory (beyond lag-1) in SPG and SubT, meaning the absence of it in the theoretical model is likely due to the use of correlation to establish the links (Born and Mignot, 2012). Physically these links could be related to the atmospheric dampening of the surface signal, which is not included.

For the proposed SubT→Rho link (red violins) models disagree on the sign and value of the instantaneous link, with a large spread for the significant models. We find a positive link at lag-1, whereas in the theoretical model the link is negative, with a cooling of SubT leading to an increase of Rho. This likely is due to the different effect of mixing on the surface density versus the density at depth. Where the mixing of the water column leads to an increase of density at depth, it actually reduces the density at the surface corresponding to the positive link found at lag-1.

The sign of the SPG $\rightarrow$ SSS link (blue violin) is very consistent in the models where it is found to be significant (but not overall). For those models it shows a negative feedback with an increase in SPG leading to a reduction in SSS a year later.

**Figure 5.** The causal effect for each of the links. The violins in colour give the distribution over the models for which the link is significant, with the black dots indicating the individual model values. The grey violins and grey dots show the non-significant results. Hatching indicates the causal effect at lag-1. The numbers at the top give the number of models for which that link is significant, respectively not significant, at the 5% level.

The Rho $\rightarrow$ SPG link (purple violins) is less consistent, but most significant models indicate that a reduction in Rho leads to a strengthening of SPG, either instantaneous or at lag-1. This is not in agreement with the theoretical mechanism, where an increase in density strengthens the gyre circulation. It indicates that the models do not capture the feedback of the density in the gyre centre to its strength. The Rho $\rightarrow$ SPG link is based on thermal wind balance (Born and Stocker, 2014), with an increase in the density difference between edge and centre of the gyre increasing the strength of the circulation. This absence could be related to the use of the barotropic streamfunction as a measure of the gyre strength, which does not fully capture the baroclinic nature of the link. It also hints at there being other processes in CMIP6 models, such as the atmospheric circulation, which act as confounding factors.

Since both links through Rho are identified with the opposite sign of the theoretical model, it is valuable to study the system when leaving out Rho, i.e. the hypothesis of a causal link from SubT to SPG. Causally the hypothesis omitting Rho is equivalent to the studied system (Pearl et al., 2016), although the effect on lag times is hard to predict. Studying this system (see Figs. S7-S9) shows large disagreement between models on the sign of the SubT $\rightarrow$ SPG link at lags of one and two years. At lag-1 slightly more significant models indicate a positive link, i.e. a cooling of SubT leading to a weakening of the SPG, which is the

opposite effect of the theoretical mechanism. However, at lag-2 slightly more models show a negative effect, in line with the hypothesis. Thus, the models do not agree on the SubT $\rightarrow$ SPG link, whether taking into account the step through Rho or not.

There is one model, CESM2, for which each link is significant either instantaneous or at lag-1. For the SSS $\rightarrow$ MLD $\rightarrow$ SubT links CESM2 indicates significant instantaneous links, whereas for SubT $\rightarrow$ Rho $\rightarrow$ SPG the significant links are found at lag-1. The sign of the links to and from Rho both have the opposite sign to the theoretical mechanism, which retains the hypothesized negative feedback loop. When omitting Rho, CESM2 has a significant negative SubT $\rightarrow$ SPG link at lag-2, being in line with the negative feedback loop.

If we do not take into account the significance of the links, but only consider the causal effect, four other models have the same sign as CESM2 for all links; CESM-FV2, CESM2-WACCM, MRI-ESM2-0 and NorESM2-LM. The models in which Swingedouw et al. (2021) found abrupt temperature shifts in the SPG region are CESM2-WACCM, MRI-ESM2-0 and NorESM2-LM. This identifies a clear connection between the presence of mechanism (B) and the possibility for abrupt shifts in CMIP6 models. That not all links in these models are found to be significant is likely linked to internal variability obscuring the signal. The key links which set this set of models apart are those to and from the SPG strength. When constraining on these three links only, i.e. negative SPG $\rightarrow$ SSS and Rho $\rightarrow$ SPG positive at lag-0 and negative at lag-1, we find this set of five models together with ACCESS-CM2. In this last model the SubT $\rightarrow$ Rho link at lag-1 is negative instead of positive, and it is unclear what this means for the possibility of an abrupt shift in the SPG as this model was not considered in Swingedouw et al. (2021).

# 5 Conclusions

We have studied the mechanisms of SPG tipping behaviour in CMIP6 models. We started with considering the local mechanisms of convection (A) (Figure 1), for which we found that, in line with observations, either an increase of salinity or a decrease of temperature causes convection (mechanism  $(A_1)$ ), i.e. an increase in mixed layer depth. Here, the effect of salinity is found to be slightly stronger than that of temperature. In contrast to the clear representation of these causes of convection, the feedback of mixed layer depth to the sea surface temperature is not identified correctly in CMIP6 models (mechanism  $(A_2)$ ). This is due to the lack of causal sufficiency of the tested hypothesis, with common drivers of temperature and salinity not being included. There are several candidates for common drivers, for example the ocean circulation (e.g. Kim et al., 2021) or the state of the atmosphere (e.g. Khatri et al., 2022), where these in turn also interact with each other. Therefore, we turned to testing mechanism (B), which is causally sufficient. This mechanism, proposed by Born and Mignot (2012), can describe bistability, and thus tipping, of subpolar gyre convection. It includes a feedback between salinity and the gyre circulation, which is in part due to eddies. Instead of focusing on the location where individual models show convection, in this part of the analysis we consider only the Labrador Sea.

We verified whether the links between the variables in mechanism (B) are well represented in CMIP6 models. The proposed mechanism, shown in Figure 1, contains a negative feedback loop on short timescales and a positive one on longer timescales. The interaction between salinity, mixed layer depth and the subsurface temperature is relatively well captured in many CMIP6 models. This shows that convection is indeed driven by surface density and is in line with the results for the local convection

mechanism  $(A_1)$ . The feedbacks between the subsurface temperature, density and strength of the subpolar gyre are present in fewer models, often with conflicting signs between models. The positive link from the temperature at depth to density at the surface can physically be explained by the upward mixing of lower density water. The mostly negative link from density to the strength of the gyre is not in line with the theoretical mechanism, indicating that models do not contain the expected interaction between density and the gyre circulation. One reason for not identifying all links is due to some models not showing (deep) convection in the Labrador Sea, in particular the CanESM5 and EC-Earth models. However, even for models with a link being significant we do not identify all links of mechanism (B). In a subset of models we identified the expected links to and from the subpolar gyre strength, although only in CESM2 these links are all significant. This subset of models includes those models in which abrupt temperature shifts in the subpolar gyre region have been identified (Swingedouw et al., 2021), indicating that this mechanism likely plays a crucial role in abrupt shifts in the subpolar gyre.

The lagged feedback from the gyre circulation to salinity is not found in any of the models and as a consequence the delayed positive feedback loop is missing. The long memory of the circulation strength can (partly) explain this, as lagged regression (used in Born and Mignot (2012)) can lead to identifying too many significant relationships (McGraw and Barnes, 2018). At a lag of two years we find a weakening effect of the circulation strength on itself in most significant models (shown in Fig. S6), which would act to increase the salinity on a timescale of around 3 years, i.e. a positive feedback loop. This is a shorter timescale than identified by Born and Mignot (2012), which can be due to the gyre-memory biasing their analysis. The gyre-memory can be explained by the transport of waters from outside the Labrador sea, which retains similar properties for multiple years, as seen during the Great Salinity Anomalies (Gelderloos et al., 2012). In CESM2 this lagged link of the gyre strength is found to be significant, providing evidence for both a negative and delayed positive feedback loop being present in this model. Out of the set of models that correctly represent the links to the gyre-strength, also CESM-FV2 and MRI-ESM2-0 show a lagged gyre-strength link, but CESM2-WACCM and NorESM2-LM do not. This does not necessarily mean that the links are not present here, because noise may have reduced their significance.

The link that contradicts most with the bistability mechanism (B) is that from density to the gyre strength. This is one of the links that relies heavily on a good parametrization of heat and salt transport by ocean eddies, and hence it is not surprising that this interaction is not identified clearly. All models considered here have a too coarse resolution to explicitly resolve ocean eddies, and hence we did not find a discernible effect of model resolution on the identification and strength of this link. Another partial explanation is that the barotropic streamfunction, which is used as a measure of the strength of the subpolar gyre, is not the most suitable variable to use for the interaction. The streamfunction is computed by vertical integration, which removes baroclinic effects, whereas it is exactly these baroclinic effects that are relevant for the positive feedback loop (Born and Stocker, 2014). Together, these likely explain the absence of this interaction in most CMIP6 models.

Despite the limitation, it is encouraging to see that the models that best represent mechanism (B) are also the models in which abrupt shifts in the subpolar gyre region are found (Swingedouw et al., 2021). It is important to better understand what sets these models apart, to improve our knowledge about potential bistability in the subpolar gyre and the likelihood of abrupt shifts in this region. Here, exploring the mechanism in eddy-rich models would be a valuable direction for future research to see whether this improves the link between density and the subpolar gyre strength. Furthermore, investigating these mechanisms in

an ocean-only model could clarify the relevance of the atmosphere for these interactions, although here one would have to be mindful of the applied atmospheric forcing. Another way to study this would be to include e.g. the North Atlantic Oscillation as a variable forcing the surface variables in mechanism (B) to see the impact this has on the causal links. Furthermore, future research could look at the impact of global warming on the variability mechanism.

The PCMCI method for causal inference used in this study has proven itself suitable to climate model datasets (Kretschmer et al., 2016; Di Capua et al., 2020; Pfleiderer et al., 2020), but it is not the only one that can be used to study causality using time-series data. Alternative methods are transfer entropy (Schreiber, 2000), conditional mutual information (Paluš et al., 2001) or convergent cross mapping (Sugihara et al., 2012; Van Nes et al., 2015). Docquier et al. (2024) compared the PCMCI-method used here with the Liang-Kleeman information flow (Liang and Kleeman, 2005), finding that both outperform standard correlation and have a comparable level of skill for the number of variables used here. With all these methods one needs to keep in mind that certain assumptions are made to arrive at the resulting causal links, and that one needs to verify these assumptions to know the results are robust. Such robustness analyses are made more complicated by e.g. lack of availability of relevant output variables in CMIP6 models.

The application of causal methods to look for mechanisms of bistability can be applied to other tipping systems when the mechanism is known (or hypothesized), as long as the hypothesized mechanism is causally sufficient. Here, one can think of the Atlantic Meridional Overturning Circulation (AMOC) or the Amazon Rainforest. Another area in which these methods can be used is the interaction between different tipping elements. Building on the results presented here, the presence of a link between the subpolar gyre and AMOC could be investigated. The subpolar gyre is one of the regions where convection takes place and links to the overturning circulation. The exact nature of this interaction and the relevance of the Labrador sea compared to other convection regions such as the Irminger or Nordic seas are still open questions. Observational data indicates that the majority of the transport related to the overturning happens through the eastern part of the subpolar Atlantic (Lozier et al., 2019). On the other hand the deepest convection occurs in the Labrador sea, indicating its relevance for bringing saline surface water to depth (Buckley et al., 2023). A recent study indicates the importance of models resolving the smaller scales to better represent the spatial heterogeneity of the ocean circulation and its relevance to convection (Gou et al., 2024). To narrow down the large uncertainty, also in model performance, causal techniques can be used to better understand which mechanisms are at play.

*Code availability.* The code used for this research is publicly available as SwindaKJ/SPG\_mechanism: SPG\_mechanism (v1.0) on Zenodo, DOI:10.5281/zenodo.13449752.

*Author contributions.* S.K.J.F. and A.v.d.H conceived the idea for this study. S.K.J.F. conducted the analysis, prepared all figures and wrote the first draft. All authors were actively involved in interpreting the results and the writing process.

Competing interests. The authors declare that they have no conflict of interest.

Acknowledgements. This publication is part of the project 'Interacting climate tipping elements: When does tipping cause tipping?' (with project number VI.C.202.081 of the NWO Talent programme) financed by the Dutch Research Council (NWO). This is ClimTip contribution #64; the ClimTip project has received funding from the European Union's Horizon Europe research and innovation programme under grant agreement No. 101137601. We acknowledge the World Climate Research Programme, which, through its Working Group on Coupled Modelling, coordinated and promoted CMIP6. We thank the climate modeling groups for producing and making available their model output,
 the Earth System Grid Federation (ESGF) for archiving the data and providing access, and the multiple funding agencies who support CMIP6 and ESGF.

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
