# Peer review of "Causal Mechanisms of Subpolar Gyre Variability in CMIP6 Models"

_EGUsphere, 2025_

## Referee Comment (RC1)

**Review of "Causal Mechanisms of Subpolar Gyre Variability in CMIP6 Models"**

**Summary of paper**

This paper explores the representation of hypothesised mechanisms that drive subpolar gyre variability in the CMIP6 piControl ensemble. The motivation is to understand how well represented the processes believed to be important for the subpolar gyre tipping point and collapse of deep convection are in models. The authors apply causal inference and causal effects approaches to show that most models only capture some elements of the important processes, and just one features all of them. This has implications for the reliability of CMIP6 models in modelling the subpolar gyre tipping point.

**Overview thoughts and recommendation**

I believe this paper makes an important contribution to understanding both the mechanisms of subpolar gyre variability, and how well models capture them. Since we are highly reliant on models to understand the risk of climate tipping points such as that of the subpolar gyre, this paper gives significant insight into how reliable these models are. Alternatively, the results could also hint at deficiencies in the popularly proposed mechanisms of subpolar gyre variability/collapse that might warrant further study. As subpolar gyre convection collapse is highly topical at present, this paper lays important groundwork for additional questions that can be addressed in future research.

I found the paper overall well written and easy to read. In particular, the use of colour coordination between the mechanisms schematic and later plots showing results was intuitive and aided understanding.

I recommend this paper for acceptance but with the following comments and questions addressed.

**Major comments**

- I am confused about the difference between the SSS to MLD pathway between the A1 and A2 mechanisms. From the schematic, it appears that that element is shared between them. However, Figure 3 makes it clear that there is a difference in the calculation for $A_1$ SSS to MLD and $A_2$ SSS to MLD. Could you elaborate on how these are different and how the calculation is done more in the text.

- Section 3: Is there any way to reformulate the hypothesised mechanisms A1 and A2 to meet the causal sufficiency condition needed for your method? For example, later you mention that atmospheric interaction has an important role – could including variables such as wind stress, E-P, latent and sensible heat help resolve this? Related to this, I wondered if there is any possible benefit to using

ocean-only simulations (e.g. OMIP) to inspect these mechanisms. Does a freely evolving atmosphere obscure some of the important processes?

- Section 4.2: "Physically these links could be related to the atmospheric dampening…". Related to my previous question, if you tested this in ocean-only experiments, would you then expect the sign to stay the same?

- Page 9: I believe there is a figure or two missing from the SI that you refer to on this page. I cannot see one relating to omitting rho and studying just SubT -> SPG, or one relating to SPG to SPG at lag-2 ("Studying this system (see SI)…" and "…circulation strength on itself in most significant models (shown in SI)…").

- You mention that CESM2 is one of the CMIP6 models that Swingedouw et al. 2021 show features an SPG convection collapse. How do the other 3 models that they find show this tipping point perform in your analysis? Are they better on average than other models at representing these mechanisms? If not, how do you reconcile this with the fact they feature a collapse of convection under future scenarios.

- You mention that mechanism B is in part driven by the role of eddies. However most CMIP6 models that you look at are too low resolution to be eddy permitting or eddy resolving. I am intrigued that you mention there was no discernible impact from model resolution on the strength of the link. What was the range of model resolutions? As far as I can tell, none of the models you look at go above $0.25°$ resolution – which is not enough to resolve eddies well. I think to really know if resolution has an important role you would need an eddy-resolving model (e.g. ~ $1/10°$ resolution or better).

- I think there are multiple promising ways to build on this work, for example looking at CESM2 in more detail to understand why it performs better, looking at high resolution models like I mention above, or including atmospheric variables in the mechanism etc. Could you add something to the discussion summarising what you think might be important next steps?

**Minor comments**

- Page 2: "Convection also feeds back to the surface as water from depth is mixed with the surface water, lowering the surface density." This statement may not be immediately intuitive. Could you add some additional descriptive words (e.g. "warmer water from depth is mixed…") to aid understanding.

- Page 2, Born and Mignot 2012 – citation needs brackets.

- Figure 1: I found labels $A_2$ and B being the same colour as one arrow in the full mechanism is a bit confusing. It implies B refers to just the green arrow for

example. Maybe move label B to a more central location and change its colour to more obviously refer to the full mechanism?

- For models where it is not output, is it possible to derive barotropic streamfunction using velocities from the model? Is there difficulty in doing this that prevented it for this study?

- Section 2.2: Python not capitalised; too instead of to ("...the longer it takes too compute...")

- Section 2.2: Can you explain in more detail what an "intervention" refers to. I did not follow why this cannot be done with time-series data.

- Section 3 (and hereafter): when you refer to the SI, can you label the particular figure you're referring to.

- Section 3: "The latter is a feature..." can you clarify if the latter refers to mechanism $A_2$ or the lag-1 memory effect? Maybe "The latter mechanism" for example will make this clearer.

- Section 4.1: "SPG -> SSS link is absent in 22 out of the 32 models". From Figure 4, I cannot see where 10 models show this link. Is that meant to be the sum of purple and orange? This does not seem to add to 10 though.

- Section 4.2: Do you have an alternative suggestion for gyre strength metric that could better capture the baroclinic nature of the gyre?

---

## Author Comment (AC1)

**Review of "Causal Mechanisms of Subpolar Gyre Variability in CMIP6 Models"**

**Summary of paper**
This paper explores the representation of hypothesised mechanisms that drive subpolar gyre variability in the CMIP6 piControl ensemble. The motivation is to understand how well represented the processes believed to be important for the subpolar gyre tipping point and collapse of deep convection are in models. The authors apply causal inference and causal effects approaches to show that most models only capture some elements of the important processes, and just one features all of them. This has implications for the reliability of CMIP6 models in modelling the subpolar gyre tipping point.

**Overview thoughts and recommendation**
I believe this paper makes an important contribution to understanding both the mechanisms of subpolar gyre variability, and how well models capture them. Since we are highly reliant on models to understand the risk of climate tipping points such as that of the subpolar gyre, this paper gives significant insight into how reliable these models are. Alternatively, the results could also hint at deficiencies in the popularly proposed mechanisms of subpolar gyre variability/collapse that might warrant further study. As subpolar gyre convection collapse is highly topical at present, this paper lays important groundwork for additional questions that can be addressed in future research.

I found the paper overall well written and easy to read. In particular, the use of colour coordination between the mechanisms schematic and later plots showing results was intuitive and aided understanding.

I recommend this paper for acceptance but with the following comments and questions addressed.

We thank the reviewer for their positive evaluation of the paper.

**Major comments**
- I am confused about the difference between the SSS to MLD pathway between the A1 and A2 mechanisms. From the schematic, it appears that that element is shared between them. However, Figure 3 makes it clear that there is a difference in the calculation for A1 SSS to MLD and A2 SSS to MLD. Could you elaborate on how these are different and how the calculation is done more in the text.

The reviewer is correct that the SSS-MLD pathway is shared between the two mechanisms. The difference that arises in Figure 3 is because of the difference in the variables on which one conditions in determining the causal effect. That is, to determine the causal effect of the link in A1 one conditions on SST, since it is also a driver of MLD, while for the causal effect in A2 this is not done as SST is no driver of MLD. We will clarify this in the text.

- Section 3: Is there any way to reformulate the hypothesised mechanisms A1 and A2 to meet the causal sufficiency condition needed for your method? For example, later you mention that atmospheric interaction has an important role – could including variables such as wind stress, E-P, latent and sensible heat help resolve this? Related to this, I wondered if there is any possible benefit to using ocean-only simulations (e.g. OMIP) to inspect these mechanisms. Does a freely evolving atmosphere obscure some of the important processes?

To fulfill the causal sufficiency condition all relevant variables need to be included. As the reviewer mentions, variables like wind stress and heat fluxes likely are drivers of both SSS and SST. The difficulty with including them in a causally sufficient network is that e.g. the wind stress can affect the heat flux and vice versa, as well as both being driven by e.g. the Jetstream, shifting the causal sufficiency condition one level up. For that reason, a "closed" hypothesis like the Born mechanism, is one that can readily be tested, but others tend to keep expanding making them unfeasible.

An ocean-only model would indeed allow studying the mechanisms in more detail, where one could use idealized atmospheric noise patterns or observations. However, to connect to the Swingedouw et al. (2021) results, the model would have to be forced by very specific atmospheric noise patterns as these are important for setting the locations where convection occurs. We will make a remark on this in the revised discussion.

- Section 4.2: "Physically these links could be related to the atmospheric dampening...". Related to my previous question, if you tested this in ocean-only experiments, would you then expect the sign to stay the same?

We do not expect these links to depend on the freely evolving atmosphere and no changes in the text needed.

- Page 9: I believe there is a figure or two missing from the SI that you refer to on this page. I cannot see one relating to omitting rho and studying just SubT -> SPG, or one relating to SPG to SPG at lag-2 ("Studying this system (see SI)..." and "...circulation strength on itself in most significant models (shown in SI)...").

We apologise for missing some figures in the SI. We will include those in the revised version.

- You mention that CESM2 is one of the CMIP6 models that Swingedouw et al. 2021 show features an SPG convection collapse. How do the other 3 models that they find show this tipping point perform in your analysis? Are they better on average than other models at representing these mechanisms? If not, how do you reconcile this with the fact they feature a collapse of convection under future scenarios.

The other models in which a collapse of SPG convection is found are CESM2-WACCM, MRI-ESM2-0 and NorESM2-LM. It could be the case that the mechanism is present in

these models does exist, but is not found to be significant for one or more links. Looking at the sign of the causal effect, representing the direction of the interaction, there is a small set of models which have the same sign as CESM2 for all links (with the constraint of the strength being at least 0.01, i.e. different from zero): CESM-FV2, **CESM2-WACCM**, **MRI-ESM2-0**, **NorESM-LM**. This is exactly the set of models in which Swingedouw et al. identified abrupt shifts (CESM-FV2 was not considered), indicating the interaction is present in these models, but obscured by internal variability. The links that are most relevant for distinguishing this set of models are those that include the SPG strength (SPG->SSS and Rho->SPG at both lags). If only constraining on these three links we find one additional model; ACCESS-CM2. This model has a different sign for the SubT->Rho links. Since it is not included in Swingedouw et al. (2021), we cannot determine how relevant this is. We will add a mention of the above in the paper and thank the reviewer for the suggestion to look into this.

- You mention that mechanism B is in part driven by the role of eddies. However most CMIP6 models that you look at are too low resolution to be eddy permitting or eddy resolving. I am intrigued that you mention there was no discernible impact from model resolution on the strength of the link. What was the range of model resolutions? As far as I can tell, none of the models you look at go above 0.25 resolution – which is not enough to resolve eddies well. I think to really know if resolution has an important role you would need an eddy-resolving model (e.g. ~ 1/10 resolution or better).

The reviewer is correct that none of the models have a high enough resolution to resolve eddies, hence the presence of the mechanism likely depends on how well they are parameterized. It would be valuable to test a model that is eddy resolving for this mechanism. We make an additional remark on this in the revised text.

- I think there are multiple promising ways to build on this work, for example looking at CESM2 in more detail to understand why it performs better, looking at high resolution models like I mention above, or including atmospheric variables in the mechanism etc. Could you add something to the discussion summarising what you think might be important next steps?

We thank the reviewer for the suggestion and will extend the discussion to include possible next steps.

**Minor comments**
- Page 2: "Convection also feeds back to the surface as water from depth is mixed with the surface water, lowering the surface density." This statement may not be immediately intuitive. Could you add some additional descriptive words (e.g. "warmer water from depth is mixed…") to aid understanding.

We will clarify this sentence to "Convection also feeds back to the surface as warmer water from depth is mixed with the surface water, lowering the surface density.".

- Page 2, Born and Mignot 2012 – citation needs brackets.

We will change this.

- Figure 1: I found labels A2 and B being the same colour as one arrow in the full mechanism is a bit confusing. It implies B refers to just the green arrow for example.
- Maybe move label B to a more central location and change its colour to more obviously refer to the full mechanism?

We have updated this figure (below), changing the colours of the labels and moving label B to a different location.

[Figure]

- For models where it is not output, is it possible to derive barotropic streamfunction using velocities from the model? Is there difficulty in doing this that prevented it for this study?

Computing the barotropic streamfunction from the velocities in the model in principle is possible. However, this requires downloading the 3D velocity fields and then integrating over those, which requires a lot of memory and computational power. For one model this would be feasible, however for doing it for all models would take a lot of computational resources and hence we refrained from it in this study.

- Section 2.2: Python not capitalised; too instead of to ("...the longer it takes too compute..."

We will correct this.

- Section 2.2: Can you explain in more detail what an "intervention" refers to. I did not follow why this cannot be done with time-series data.

With an intervention we refer to interfering in the system, like can be done when studying waves in a wave tank by changing the e.g. forcing frequency. Such an intervention allows for cleanly separating cause and effect. However, in the climate system such an intervention, for example changing the salinity in the Labrador sea is unfeasible and undesirable. In climate models such interventions are possible, but computationally very expensive and therefore unfeasible. We will clarify this point in the text.

- Section 3 (and hereafter): when you refer to the SI, can you label the particular figure you're referring to.

We will do this.

- Section 3: "The latter is a feature..." can you clarify if the latter refers to mechanism A2 or the lag-1 memory effect? Maybe "The latter mechanism" for example will make this clearer.

This refers to the lag-1 memory effect. We will clarify this.

- Section 4.1: "SPG -> SSS link is absent in 22 out of the 32 models". From Figure 4, I cannot see where 10 models show this link. Is that meant to be the sum of purple and orange? This does not seem to add to 10 though.

The number of 22 refers to the absence of the link for all lags. With 8 models showing the link at lag-1, and two more at lag-0 (next to some that have both as significant). We will add a note that this is for all lags in the text.

- Section 4.2: Do you have an alternative suggestion for gyre strength metric that could better capture the baroclinic nature of the gyre?

We also considered sea surface height as a measure of gyre strength. We found both were strongly linked, but that the links with the barotropic streamfunction were stronger. We have thought about a better metric to capture the baroclinic contribution of the gyre circulation. However, any definition would depend on the density in the gyre one way or another, meaning such a gyre metric would automatically be linked to the density (and thus temperature and salinity). This way you likely "build in" a link between e.g. rho and the SPG strength, which would complicate a proper verification of the mechanism.

---

## Author Comment (AC2)

**Review of "Causal mechanisms of SPG variability in CMIP6 models"**
This study aimed at assessing whether the hypothesized physical mechanisms that could lead to a tipping of SPG convection are effectively represented in modern climate models. To this end, the authors analyzed unforced piControl simulations from CMIP6, and evaluated those interactions that could give rise to SPG convection variability and/or bistability. They applied causal inference and causal effect estimation techniques to test the presence, strength, and direction of the commonly proposed causal links among the key variables in the SPG system. The results show that while interactions between salinity, temperature, and mixed layer depth are relatively well represented in many CMIP6 models, the feedbacks involving density and gyre strength are either inconsistently captured or missing entirely. Only one model, i.e. CESM2, exhibits full consistency with the proposed feedback mechanisms underlying SPG bistability.

**1. General assessment**
The study is highly relevant and valuable, as it tackles a crucial issue for understanding and predicting future climate change: to what extent climate models can capture the key mechanisms governing SPG dynamics. The SPG convection, commonly proposed as an independent tipping element of the climate system, has already been projected to collapse in some future projections, with severe impacts on the North Atlantic. However, the models exhibiting such a collapse do not appear to share common patterns and underlying processes. Furthermore, the SPG convection is closely linked to the AMOC, another recognized major tipping element, whose disruption could lead to even broader climatic consequences.

The comprehensive coverage of CMIP6 models allows the authors to draw robust conclusions about model spread and biases when reproducing mechanisms in the SPG. In this context, the results for CESM2 are particularly noteworthy: it is the only model that shows both a convection collapse in future projections and a consistency with the physical mechanisms underpinning SPG bistability. This finding supports the plausibility of the potential bistability of the SPG convection, thus reinforcing its potential role as a climate tipping element. Conversely, the absence of such mechanisms leading to bistability found in the other models suggests that SPG tipping, as shown in other studies focusing on future projections, might rather reflect a nonlinear and irreversible response to external forcing. Nonetheless, the lack of detected bistability mechanisms in these models may also stem from methodological limitations in the current analysis, as discussed by the authors in the text and also detailed below in the major comments. I think that these limitations do not undermine the overall quality of this study, but rather point to valuable directions for future research.

A key strength of the paper lies in its innovative application of causal inference and effect techniques, i.e. the PCMCI+ framework, to disentangle the causal structure and feedbacks within the SPG system. This represents a notable methodological advancement when investigating SPG dynamics, which traditionally relies on

correlation-based diagnostics. However, the experimental design used to analyze mechanisms A1 and A2 could possibly benefit from further refinement to match the overall rigor and quality of the paper, as detailed below in the major comments.
The figures in the manuscript are well-designed and effectively informative, while a couple of figures referenced in the SI appear to have been inadvertently omitted. The manuscript is grounded in a clear hypothesis framework, and generally follows a logical narrative. However, I find certain parts technically dense and too hard to understand at a first read. This mainly reflects the inherent complexity of both the methodology used and the SPG dynamical system, yet there is space for some simplification or improvement as detailed below.

Overall, I think this is a strong and well-executed paper that provides timely and insightful results through the use of an innovative methodology. The few limitations appear to likely arise from methodological constraints and minor omissions. With a few additional refinements and clarifications, I believe the manuscript would be very suitable for publication.

We thank the reviewer for their positive evaluation of the paper.

**2. Major comments**

- *Mechanisms A1 and A2*

I find the analysis of mechanisms A1 and A2 rather weak with respect to the overall study. This is for the following reasons:

(i) Violation of causal sufficiency: as mentioned by the authors, this analysis omits potential common drivers, which conflicts with the otherwise high level of scientific rigor in the study.

(ii) Redundancy with mechanism B: the main outcome of this part of the study, i.e. the strong causal link SSS → MLD in mechanism A1, is already robustly captured in the more comprehensive analysis of mechanism B.

(iii) Lag structure ambiguity: the methodology for estimating causal effects in A1 and A2 is not fully clear. I suppose that links have been evaluated at lag 0 (?), i.e. contemporaneous links, which raises concern for mechanism A2. Indeed, the restratification feedback (MLD → SST) would require a lag of at least one year to be detectable, as deep convection occurs in winter, while restratification starts in the following spring. The effects on SST would be therefore visible only from lag-1 on.

More broadly, I cannot understand why mechanisms A1 and A2 were treated independently. To my understanding, the PCMCI+ algorithm should be able to handle causal networks that include bi-directional links at different lags (i.e., SST → MLD at lag 0, and MLD → SST at lag 1). If the separation between A1 and A2 is due to a limitation in the PCMCI+ algorithm, it would be helpful to clarify this point in the text. Otherwise, applying PCMCI+ to a unified A1+A2 structure may yield a more consistent representation of the SPG convection feedbacks. If this is technically feasible, and in a relatively short time, I would therefore encourage the authors to consider a unified

A1+A2 structure, with also the inclusion of a shared parent for SSS and SST, e.g. by including the SPG gyre strength in this loop (or, said in a different way, by expanding the yellow box to SPG gyre strength in Fig.1). The use of alternative confounders of SST and SSS, such AMOC or NAO, would also offer valuable insights into the mechanisms of SPG convection. However, given the additional effort that such an analysis would require, it is reasonable to present this as a potential direction for future work in the Conclusions section.

The reviewer is correct that the causal inference step of PCMCI+ in principle is able to detect the links between the variables. However, when doing this the strongest link identified is one between SST and SSS at different lag times. Common drivers lead to a strong (lagged) correlation between the two, meaning the outcome of causal inference is hard to interpret. Therefore, we focussed our analysis for mechanisms A1 and A2 on the causal effect given the hypothesis of these mechanisms.

The difficulty with expanding the current analysis to include the SPG strength to have a causally sufficient mechanism is that the analysis in mechanisms A1 and A2 is on the local processes, with the location being different between all models (whereas the analysis for mechanism B is for the Labrador sea). This means that the inclusion of the SPG strength as a common driver would give results that are hard to interpret given the different locations that are considered (see Fig. 2) and therefore we refrain from doing so. The different locations used to study the different mechanisms also means that the A1 and A2 analysis is not redundant, as it allows for better identifying the local convection dynamics. For example, the signal of the SSS->MLD link is significantly stronger when considering the model-specific convection locations.

We will clarify the above reasoning on the distinction between local and Labrador sea processes in the text. In addition we will better describe the lag structure used for both mechanisms. The reviewer is correct that all links here are contemporaneous. We will add a discussion of lags for mechanism A2 and thank the reviewer for noting this likely is more relevant for the MLD->SST link at lag-1. We have now computed these, showing an improvement on the identified restratification feedback as seen in the plot below, and will add the results to the revised manuscript.

[Figure]

• *The sensitivity of causal links to the model-specific sites of deep convection*
Concerning the mechanism B, results show a generally weak causal connection
between SubT, rho, and SPG (gyre strength) across most CMIP6 models. This is an
important and intriguing finding, in apparent contradiction with proposed mechanisms
of bistability of the SPG convection. The authors explained these outcomes as
associated with the use of the barotropic flow rather than baroclinic as a measure of
SPG (gyre strength), and with the role of other confounding factors not considered here,
e.g. the atmospheric circulation. While these interpretations are physically plausible, I
am concerned that the result may more merely reflect limitations of the experimental
design. Specifically, for the analysis of mechanism B, the authors considered a fixed
spatial box over the Labrador Sea. However, in at least 10 out of 32 models, convection
may be substantially weaker or even absent in this region. This spatial mismatch, i.e.
the different location of the major convection sites among models, could severely
impact the estimation of causal relationships in the fixed box, and introduce large inter-
model spread, as indeed seen in Figure 5, notably for SubT→rho and rho→SPG. I believe
this caveat warrants explicit discussion in the manuscript.

In this context, it would also be helpful to clarify which depth levels of potential
temperature (thetao) were used to compute SubT. Given the varying convection depths
across models, this information is essential for assessing the robustness and
comparability of the results. Also, it is not clear to me if the "density at the centre of the
SPG" is the surface density averaged over the entire Labrador box or it has been defined
in a different way.

The reviewer is correct in noting that the choice of the Labrador sea box is one of the
limitations of this study. For example, the CanESM-1 model, in which no significant

links are identified, does rarely show deep convection occurring in the Labrador sea (instead it occurs in the Nordic seas). We will add some discussion on this in the manuscript. In addition we will clarify the depth levels considered (50-1000m depth). The reviewer is correct that the density is computed over the entire Labrador sea box.

- *General readability*

The manuscript is very well written, yet a few minor revisions could further improve the clarity and flow of the text. First, references to figures in the SI should be more precise, ideally citing the specific figure numbers rather than using a too generic "see SI." Note that several supplementary figures referenced in the main text appear actually to be missing from the SI. Please fix it!

We apologise for part of the SI missing and will correct this. We will also add references to the exact figures and tables in the SI.

Also, I recommend reconsidering the use of the term "direct link" as a synonym for "contemporaneous link" (i.e., causal connections at lag-0, which can be "directed" or "unoriented"). This terminology may lead to confusion, as "directed link" in causal graphs refers more broadly to the presence of a causal direction, regardless of the lag. This apparently subtle issue led me to some initial misinterpretation of the text, which can be easily avoided by adopting a less ambiguous term.

We thank the reviewer for noting this and will consistently use contemporaneous link throughout.

**3. Specific minor comments and technical corrections**

Page 3 – Fig. 1 → why the potential link "SPG strength" to "SST in SPG" has not been considered for the analysis of mechanism B? Also, I suggest to clarify in the caption the reason for framing SST with dashed lines. Without instructions, I assume this is intended to highlight the dual role of SST within the mechanisms, as it acts as a parent in A1 and a child in A2. Is it correct?

Mechanism B is a hypothesis is based on the study of Born, in which SST was not found to be of strong relevance for the mechanism. Therefore, we do not include it here.

The reason for the dashed lines around SST is to indicate it is only included in mechanisms A1 and A2, and not in mechanism B. We will clarify this in the caption

Page 5 – Fig. 6 → are the causal effects between variables in mechanisms A1 and A2 estimated at lag 0 (i.e., contemporaneous), while accounting for persistence through lag-1 autocorrelation? Please provide more details on this point.

The reviewer is correct that they are estimated at lag-0 with lag-1 autocorrelation. We will clarify this in the manuscript.

Page 5 – "Secondly, the time series is assumed to be stationary, in the sense that the presence and strength of the links does not change in time." → instead of referring to "stationary time series" I would rather express this as a "stationary set of causal links".

We thank the reviewer for the suggestion and will change this phrasing.

Page 6 – "The number of models for which each of the five links is significant are shown in Figure 4 for lags up to 10 years (individual model results are shown in the SI)." → this is not shown in SI.

We will correct the SI and ensure all figures are included.

Page 7 – "All but one model capture at least one step of the process (B) (only CanESM5-1 does not)." → it would be interesting to know more about the MLD pattern in this model, particularly the location of its maximum relative to the Labrador Sea box. Unfortunately, Fig. 2 does not allow to identify it, likely because "not all markers are visible due to overlapping maxima." Could the inability of this model in capturing any step of the mechanism B be related to the specific MLD pattern? Clarifying this point could possibly help interpreting the inter-model spread.

The location of maximum MLD in CanESM5-1 is located in exactly the same spot as that in CanESM5, south of Iceland. Both CanESM5 models are among those that show the least convection in the Labrador sea, instead their convection happens outside the SPG region in the Nordic seas. We will add a note of this in the manuscript.

Page 7 – "The links to and from SPG are found in only 10 and 17 models respectively." → I think that it is not "SPG" here but rather "rho".

We thank the reviewer for noting this mistake and will double check the numbers.

Page 8 – "In Figure 5 the causal effect of each link is shown for models with and without that link being significant following the causal discovery step (values for individual models are given in the SI)." → this figure in SI is actually missing.

We will correct the SI and ensure all figures are included.

Page 8 – "At lag-1 both these links are found with the opposite sign in most models where they are significant, contrary to the theoretical model." → here "both these links" means SSS to MLD and MLD to SubT? Please specify it.

We will specify this in the manuscript.

Page 9 – "Studying this system (see SI) shows large disagreement between models on the sign of the SubT→SPG link at lags of one and two years." → this is actually not shown in SI.

We will correct the SI and ensure all figures are included.

Page 10 – "The absence of the interaction between density and gyre strength means that we cannot be certain of the existence of the bistability mechanism in many CMIP6 models, and thus whether they are able to exhibit tipping behaviour in SPG convection." → this phrasing seems to suggest that tipping behaviour necessarily depends on the presence of bistability, which is not always the case. While bistability does allow for tipping through bifurcation, a tipping point more generally refers to a critical threshold at which a small change in forcing leads to a rapid shift in the state of a dynamical system. Such behaviour can also occur without bistability, e.g., through strongly nonlinear but irreversible responses. I suggest to rephrase this part.

We thank the reviewer in noting that we implicitly use a definition of tipping that is linked to bistability. We will clarify the link between tipping and SPG bistability based on the mechanistic understanding in the introduction. Furthermore, we will rephrase this part to be in line with the added clarification.

Page 10 → (Gou et al., 2024) should be (Gu et al., 2024).

We will correct this.

Page 10 → "grye strength" should be "gyre strength".

We will correct this.

Page 14 → "Tippint" should be "Tipping"; "Jonathen" should be "Jonathan".

We will correct this.